# Comparison of Muscle Regeneration Effects at Different Melittin Concentrations in Rabbit Atrophied Muscle

**DOI:** 10.3390/ijms25095035

**Published:** 2024-05-05

**Authors:** Byeong-Churl Jang, Eun Sang Kwon, Yoon-Jin Lee, Jae Ik Jung, Yong Suk Moon, Dong Rak Kwon

**Affiliations:** 1Department of Molecular Medicine, College of Medicine, Keimyung University, Daegu 42601, Republic of Korea; jangbc123@gw.kmu.ac.kr; 2Department of Medicine, College of Medicine, Keimyung University, Daegu 42601, Republic of Korea; 3Department of Biochemistry, College of Medicine, Soonchunhyang University, Cheonan 31151, Republic of Korea; leeyj@sch.ac.kr; 4Department of Rehabilitation Medicine, Catholic University of Daegu School of Medicine, Daegu 42472, Republic of Korea; smart03058@naver.com; 5Department of Anatomy, Catholic University of Daegu School of Medicine, Daegu 42472, Republic of Korea; ysmoon@cu.ac.kr

**Keywords:** muscle atrophy, Melittin, muscle regeneration, concentration, rabbit models

## Abstract

This research aimed to explore the healing impacts of Melittin treatment on gastrocnemius muscle wasting caused by immobilization with a cast in rabbits. Twenty-four rabbits were randomly allocated to four groups. The procedures included different injections: 0.2 mL of normal saline to Group 1 (G1-NS); 4 μg/kg of Melittin to Group 2 (G2-4 μg/kg Melittin); 20 μg/kg of Melittin to Group 3 (G3-20 μg/kg Melittin); and 100 μg/kg of Melittin to Group 4 (G4-100 μg/kg Melittin). Ultrasound was used to guide the injections into the rabbits’ atrophied calf muscles following two weeks of immobilization via casting. Clinical measurements, including the length of the calf, the compound muscle action potential (CMAP) of the tibial nerve, and the gastrocnemius muscle thickness, were assessed. Additionally, cross-sectional slices of gastrocnemius muscle fibers were examined, and immunohistochemistry and Western blot analyses were performed following two weeks of therapy. The mean regenerative changes, as indicated by clinical parameters, in Group 4 were significantly more pronounced than in the other groups (*p* < 0.05). Furthermore, the cross-sectional area of the gastrocnemius muscle fibers and immunohistochemical indicators in Group 4 exceeded those in the remaining groups (*p* < 0.05). Western blot analysis also showed a more significant presence of anti-inflammatory and angiogenic cytokines in Group 4 compared to the others (*p* < 0.05). Melittin therapy at a higher dosage can more efficiently activate regeneration in atrophied gastrocnemius muscle compared to lower doses of Melittin or normal saline.

## 1. Introduction

Muscle dysfunction or atrophy significantly diminishes quality of life and interferes with the daily routine of patients. Long periods of bed rest or lack of movement can cause muscle atrophy, resulting in substantial neuromuscular deconditioning [1]. The adverse health effects of immobilization include reduced myofibrillar protein levels, disturbed metabolic processes, changes in the blood and nervous systems, and the facilitated irregular buildup of ectopic fat within muscles [2].

Upon muscle injury, the division of satellite cells, which are myogenic precursor cells, is activated. These cells are an integral part of repairing and regenerative processes in skeletal muscles [3]. Nonetheless, the proliferation capacity of these cells in the muscles can be affected by external force and a lack of mobility [4].

Emerging research indicates that the muscle injury and atrophy caused by immobilization initiate a chronic inflammatory response, marked by increased levels of pro-inflammatory biomarkers [3,4]. This response is further amplified by cytokines like TNF-α and IL-6, which are abundant in inflammation and accelerate muscle atrophy by promoting the ubiquitin–proteasome pathway, causing muscle degradation and deteriorated functioning. Additionally, oxidative stress has been proven to be a crucial factor impairing the balance between protein formation and elimination in the progression of immobilization-provoked muscle atrophy [5,6]. High levels of reactive oxidative stress induce muscle atrophy through the promotion of NF-κB pro-inflammatory expression [7]. These insights suggest that treatments involving anti-inflammatory substances and antioxidants may be effective in combating the muscle atrophy resulting from immobilization.

Bee venom is extensively utilized as a treatment for arthritis and rheumatoid conditions, particularly in East Asian nations like Korea, China, and Japan. Melittin, a primary constituent of bee venom, is a peptide consisting of 26 amino acids and is characterized by the capability to create channels in cell membranes [8]. Several studies have evaluated the biological and pharmacological features of Melittin, showing that it has numerous potential therapeutic applications, including antiviral, antibacterial, antifungal, antiparasitic, anticancer, and neuroprotective effects [9]. Furthermore, Melittin therapy has been found to have pain-relieving effects, anti-rheumatoid arthritis properties, and immune-modulatory activity [10]. Based on the various biological activities of Melittin, it has been developed as a treatment for a broad spectrum of diseases covering inflammatory conditions like osteo- and rheumatoid arthritis along with neurological disorders including amyotrophic lateral sclerosis (ALS) and Parkinson’s disease (PD) [11].

However, the possible beneficial impact of Melittin on muscle regeneration has not been extensively explored. The remarkable anti-inflammatory effect of Melittin is expected to treat inflammatory responses in atrophied muscle. Additionally, it is expected to regenerate atrophied muscle induced by immobilization. Therefore, our research was designed to assess the anti-inflammatory and regenerative properties of Melittin on atrophied calf muscle in cast-immobilized rabbits.

## 2. Results

No significant variance was detected in the average percentage of muscle atrophy among the four groups when evaluating clinical, imaging, and electrophysiologic indicators after two weeks of immobilization (*p* < 0.05, Table 1, Figure 1). Groups 3 and 4 exhibited significantly greater average percentages of regenerative changes in the thickness of the right medial and lateral GCMs, the circumference of the right calf, and the CMAP amplitude of a right tibial nerve compared to Group 1 (*p* < 0.05, Table 2, Figure 2). While there were no significant disparities in these parameters between Groups 3 and 4, there was a tendency for higher regeneration efficacy with increasing Melittin dosage.

The histological parameters did not differ significantly among the four groups (*p* < 0.05, Table 3, Figure 3). Group 2, Group 3, and Group 4 were all characterized by significantly greater average CSAs of type I and II medial and lateral GCM muscle fibers than Group 1. These parameters in Group 4 had the largest values among all groups.

Immunohistochemistry revealed significant differences between Group 1 and the others (*p* < 0.05, Table 3). As shown in Table 3, Melittin significantly increased the VEGF, PECAM-1, PCNA, and BrdU ratios in the medial and lateral GCM muscle fibers at all three assessed doses (4, 20, or 100 μg/kg) in a dose-dependent manner.

The VEGF ratios of the two GCM fibers in Group 4 significantly exceeded those in the remaining three groups. The values for these parameters were significantly greater in Group 2 and Group 3 than in Group 1 (*p* < 0.05, Table 3, Figure 4). Similarly, the PECAM-1 ratios of the two GCM muscle fibers were significantly higher in Group 2, Group 3, and Group 4 than in Group 1 (*p* < 0.05, Table 3, Figure 4). The PECAM-1 expression of both GCM muscle fibers did not differ significantly between Groups 3 and 4. The PCNA ratios of lateral GCM muscle fibers and the BrdU ratios of both GCM muscle fibers in Group 4 were significantly greater than in the other groups (*p* < 0.05, Figure 5). The PCNA and BrdU ratios of the two GCM muscle fibers in Groups 2 and 3 significantly exceeded that in Group 1 (*p* < 0.05, Table 3, Figure 5). The PCNA and BrdU ratios of the two GCM muscle fibers in Group 3 significantly surpassed those in Group 2 (*p* < 0.05, Figure 5).

The densities of VEGF expression of the GCM muscle fibers on both sides, as measured by Western blotting, had significantly higher values in Groups 2–4 than in Group 1 (*p* < 0.05, Figure 6). The VEGF expression had a significantly higher density in Group 4 than in the other groups. The protein levels of VEGF in atrophied muscle correlated positively with the doses of Melittin.

Group 4 revealed significantly lower expression levels of IL-1α, IL-1β, TNF-α, and p38 than the other three groups (*p* < 0.05, Figure 6 and Figure 7). Group 4 revealed significantly higher expression levels of PCNA and Akt than the other three groups (*p* < 0.05, Figure 7). TNF-α expression in the lateral GCM muscle fibers did not differ significantly between Group 2 and Group 3. Group 4 did not display significantly different expression levels of IL-1α, IL-1β, or VEGF on GCM muscle fibers on either side compared with the other groups.

## 3. Discussion

A key discovery from this study is the enhanced regenerative impact on the atrophied GCM in groups treated with Melittin versus those treated with normal saline. Treatment with a relatively higher dose of Melittin (100 μg/kg) was more effective in mitigating the negative outcomes of a two-week immobilization casting (IC) compared to either the absence of it or its administration at lower doses (4 μg/kg or 20 μg/kg). This finding is corroborated by both clinical and electrophysiological evaluations (such as measurements of calf circumference, the compound muscle action potential of the tibial nerve, and the thickness of GCM via ultrasound), as well as histological assessments (CSA of muscle fibers).

Muscle atrophy is strongly linked to an increase in the generation of inflammation-inducing cytokines, including TNF-α (tumor necrosis factor-alpha), IL-1 (interleukin-1), IL-6 (interleukin-6), and IFN-γ (interferon-gamma) [12]. Patients with chronic conditions, such as kidney or heart failure, often exhibit elevated levels of TNF-α and its soluble receptors in their bloodstream [13,14]. Similarly, cancer patients experiencing cachexia display elevated levels of various inflammation-promoting cytokines contributing to the ongoing muscle mass reduction seen in these diseases [15]. Exposure to pro-inflammatory cytokines like TNF-α, IL-1, or IL-6 in healthy animals leads to muscle degradation, characterized by increased ubiquitin expression and proteasome activity [16,17]. Chronic or excessive inflammation can impede recovery processes and cause further damage to skeletal muscle, highlighting the importance of inflammation management in recovering from muscle atrophy. This study has demonstrated that Melittin therapy effectively reduces the levels of the pro-inflammatory cytokines IL-1α, IL-1β, and TNF-α, with a dose-dependent anti-inflammatory effect observed in the Western blot analysis. Additionally, Melittin treatment significantly enhanced the CMAP of a tibial nerve in atrophied GCMs, indicating Melittin’s neuroprotective properties. Previous research has shown Melittin’s anti-inflammatory actions in cases of cerebral ischemia, where it reduced the elevated cytokine levels (IL-1β, IL-6, TNF-α) post-ischemia by inhibiting the NF-κB signaling pathway linked to inflammatory factor production [18]. Similarly, Melittin treatment in models of amyotrophic lateral sclerosis (ALS) has been shown to improve motor function and decrease neuroinflammation and neuron death in the spinal cord, as opposed to control groups [19].

Immunohistochemical staining identified a notable increase in VEGF-positive cells and a higher microvascular density marked by PECAM-1 in Group 4 than in the other groups. This suggests that Melittin’s stimulation of VEGF and PECAM-1 could potentially boost regenerative outcomes. VEGF plays a critical role in encouraging the proliferation of endothelial cells, initiating angiogenesis, and managing the production of collagen fibers [20]. PECAM-1, recognized for its role in angiogenesis, is crucial for healing and regeneration processes [21]. These insights are in line with research conducted by Badr et al. [22], which showed that Melittin enhances the levels of TGF-β (tissue growth factor-beta) and VEGF, facilitating wound healing in diabetic rats through the promotion of angiogenesis and collagen creation. Similarly, Lee et al. [23] observed significant recovery in an injured biceps femoris muscle of a Melittin-treated mouse model, owing to the reduction in inflammation-inducing cytokines and an increase in muscle regeneration indicators like MyoD (myoblast determination protein 1), Myogenin, and α-SMA (alpha-smooth muscle actin). Consistent with these findings, our study indicates that Melittin treatment elevates the levels of PCNA and BrdU, markers associated with cell growth and DNA replication.

While certain studies propose that Melittin can elevate the expression of VEGF and PCNA, counter-evidence suggests it might actually reduce their levels. The compound has demonstrated significant anti-cancer capabilities through mechanisms like inducing apoptosis, arresting the cell cycle, altering oncogenic signaling pathways, and hindering metastasis. Melittin has been shown to diminish VEGF and PCNA expression in cancer cells while sparing physiologically normal cells [24]. In lung cancer mouse models, Melittin achieved tumor suppression by selectively targeting and reducing VEGF, PCNA, and CD31 (an angiogenesis marker) within the tumor cells [25,26]. The effects of Melittin on VEGF and PCNA expressions seem to depend on variables such as the type of cells involved, Melittin dosage, and the particular experimental setup. This indicates the necessity for further studies to clarify how Melittin influences these factors and to fully understand its therapeutic potential and mechanisms of action.

In our study, we observed that Melittin administration dose-dependently modulates key signaling pathways involved in muscle regeneration. Specifically, we found a dose-dependent decrease in the activity of the p38 MAP kinase pathway and an increase in the Akt pathway activity. The p38 pathway is typically associated with the stress response and inflammation, and its downregulation in our model aligns with the reduced inflammatory response, which is beneficial for muscle recovery [27]. On the other hand, the activation of the Akt pathway is crucial for promoting cell survival, growth, and proliferation, which are essential processes in muscle regeneration [27]. These findings are significant because they provide a mechanistic explanation for the beneficial effects of Melittin observed in our model of immobilization-induced muscle atrophy. By modulating these pathways, Melittin facilitates a more conducive environment for muscle repair and regeneration, suggesting its potential as a therapeutic agent in conditions characterized by muscle atrophy.

The safety of Melittin in clinical settings remains a significant concern, as it is capable of causing hemolysis. This could potentially cause rhabdomyolysis with subsequent acute kidney failure, presenting a wide spectrum of clinical manifestations from local edematous skin swelling to severe systemic anaphylactic shock. To mitigate these risks, several strategies have been elaborated, such as the use of nanotechnology, gene therapy, and immunoconjugates applied to Melittin [28,29].

A previous study conducted a 13-week toxicity evaluation on Sprague–Dawley rats using Sweet Bee venom delivered via intramuscular injections. The results demonstrated that doses up to 0.07 mg/kg, administered into the femoral muscle, were safe and did not cause any harmful effects [30]. Extending these findings, our research indicates that doses as high as 210 µg/kg are safe in 3 kg rabbits, without eliciting adverse reactions. Based on these insights, we recommend using melittin at various dosages: low (4 µg/kg), medium (20 µg/kg), and high (100 µg/kg). In addition, an acute toxicity test on male adult BALB/c mice, aged 8 weeks and weighing between 20 and 25 g, established the LD50 for Melittin at 4.95 mg/kg, with the maximum sub-lethal dose at 2.4 mg/kg [31]. Our studies confirmed no toxicological impacts from Melittin, even at the highest concentration of 100 µg/kg tested. Furthermore, prior studies using intraperitoneal administration in mice (at doses of 0.75 and 1.5 mg/kg) and rats (at 25, 50, and 100 µg/kg) showed no significant changes in immune parameters, underlining its immunological safety [32,33]. Notably, the highest dose in rat studies, 100 µg/kg, matches the high concentration used in our research. We did not observe any adverse or untoward events.

The present study has some limitations along with the considerations for future studies. First, most of the research on Melittin’s effects, including the present study, has been conducted in preclinical studies using animal models or cell cultures. Further research, particularly in human studies, is necessary to fully understand the effects of Melittin. Second, the assessment period was restricted to 4 weeks, which included baseline measurements, 2 weeks of immobilization, and Melittin injections. Thus, further research is required to evaluate the long-term effects of Melittin injections on atrophied muscles. Third, three different dosages of Melittin (4 μg/kg, 20 μg/kg, and 100 μg/kg) were used. The potential variations in the efficacy of Melittin therapy at different dosages were not analyzed in this research. Future work should investigate more relevant dosages of Melittin for atrophied muscles. Last, with regard to Melittin’s impact on normal muscle tissue, our study did not directly investigate this aspect. However, based on the safety profile observed in our and prior studies, we propose that future research should explore Melittin’s effects on normal muscle to further validate its safety and efficacy.

## 4. Methods

### 4.1. Animal Grouping

The approval for the research protocol was received from the Institutional Animal Care and Use Committee (IACUC) of the Catholic University of Daegu School of Medicine, adhering to the IACUC guidelines for animal care and use (IRB No. DCIAFCR-230512-07-Y). Male New Zealand white rabbits (n = 24, 12 weeks old) with an average weight of 3.3 kg (ranging from 2.8 to 3.6 kg) were housed individually in steel cages constantly preserved at 23 ± 2 °C and under humidity conditions of 45 ± 10%. The animals were fully supplied with tap water and a commercial rabbit diet. After 1 week of adaptation, the rabbits were randomly divided into four groups (n = 6 per group). Following two weeks of the immobilization of a right leg using a cast (IC), the cast was removed, and the hair from the lower extremities of the animals was eliminated with a commercially available device. Subsequently, the four groups underwent different procedures on the right atrophied calf muscles (Figure 8): Group 1 had weekly injections of 0.2 mL normal saline (G1-NS), Group 2 received 4 μg/kg Melittin (G2-4 μg/kg Melittin), Group 3 received 20 μg/kg Melittin (G3-20 μg/kg Melittin), and Group 4 received 100 μg/kg Melittin (G4-100 μg/kg Melittin) for two weeks. Ultrasound-guided injections were administered in accordance with physiatrists’ standards. A commercially available ultrasound system with a 5–18 MHz multifrequency linear transducer (EPIQ 5; Philips Healthcare, Andover, MA, USA) was used for the injections. No medication was administered to the rabbits, and all were sacrificed two weeks after the first injection.

### 4.2. Immobilized by Cast (IC)

For 2 weeks, the right gastrocnemius muscle (GCM) of the rabbits was stabilized with a PVC plastic splint, a non-adhesive bandage, and an adhesive elastic bandage (Tensoplast^®^; Smith & Nephew Medical, London, UK) according to standardized IC techniques [34]. We extended the right knee and ankle using a splint (Figure 9).

### 4.3. Injection Procedures

All injections were administered intramuscularly under regional anesthesia with Zoletil^®^ 50 (15 mg/kg, Virbac Korea, Seoul, Republic of Korea) and xylazine (5 mg/kg, Rompun^®^; Bayer Co., Seoul, Republic of Korea). Normal saline or Melittin (0.1 mL each) were given at two sites—both sides of the GCM muscle—with ultrasound guidance using a 5–13-MHz multifrequency linear transducer (Antares; Siemens Healthcare, Erlangen, Germany) (Figure 10). Commercially obtained Melittin (Rural Development Administration, Jeonju, Republic of Korea) diluted with 0.9% normal saline was used.

The solution (0.1 mL) was administered into both the lateral and medial sides, aligned horizontally, guided by a central reference point. This was established at the midpoint between two specific locations: one being the point at the proximal third of a line marked from the midpoint between the malleoli of the ankles to between the femoral epicondyles, and the other being the medial and lateral extremities of a horizontal line drawn at right angles to this longitudinal line. The treatment process was repeated at the same locations one week after the first injection.

### 4.4. Clinical Parameters

A physiatrist, unaware of the group assignments, conducted all measurements for the study. To evaluate the amplitude of the CMAP in a tibial nerve, a motor nerve conduction study was carried out. The active electrode was positioned at the center of a GCM, and the reference was placed on the subcutaneous tissue near the ankle. Electrical impulses were transferred to a nerve at the popliteal fossa, and the peak CMAP response was recorded after 7–10 supramaximal stimuli.

The greatest length measured using a tape around circumference of the calf regions was determined, with the rabbits’ knee joints bent at a 90-degree angle and the ankles relaxed. The thickness of both sides of the GCM in the longitudinal plane was evaluated from the outermost layer to deep fascia using real-time B-mode ultrasound imaging (Figure 11). The pictures of the GCM were captured at the predetermined points corresponding to the injection sites on the mentioned sides of the muscle.

Finally, regenerative changes in CMAP amplitude, GCM thickness, and calf circumference were evaluated with the following equation: [(Values measured two weeks after first injection − Values measured after two weeks of IC)/Values measured two weeks after first injection × 100]. The results were presented as a percentage of the changes observed on the right side.

### 4.5. Tissue Preparation

After completing all intramuscular injections, the rabbits were euthanized under general anesthesia. Both the medial and lateral sections of the GCM from all rabbits were carefully eliminated and preserved in neutral-buffered formalin for 24 h. Following this preservation step, the tissue samples were encased in paraffin (Paraplast; Oxford, St. Louis, MO, USA) and then precisely serially sectioned in the transverse plane, each with a thickness of 5 μm, to prepare for detailed examination.

### 4.6. Immunohistochemistry

The slices underwent staining for type I and type II fibers using monoclonal anti-myosin antibodies for Skeletal, Slow (Sigma-Aldrich, St. Louis, MO, USA) and Skeletal, Fast (Sigma-Aldrich), respectively (Figure 12). To identify cells undergoing proliferation within the sections, immunostaining was performed with monoclonal anti-PCNA (proliferating cell nuclear antigen) antibody (clone PC10; Santa Cruz Technologies, Dallas, TX, USA), and monoclonal anti-BrdU (bromodeoxyuridine) antibody (clone BU-33; Sigma-Aldrich). Angiogenic markers were then detected using polyclonal antibodies against VEGF (vascular endothelial growth factor) (A-20; Santa Cruz Biotechnology, Santa Cruz, CA, USA) and PECAM-1 (platelet endothelial cell adhesion molecule-1) (M-20; Santa Cruz Biotechnology).

All rabbits were injected intraperitoneally with BrdU (25 mg/kg, B5002; Sigma-Aldrich) for the purpose of staining. A day after the BrdU was given, the rabbits were euthanized, and their tissues were divided into paraffin-embedded sections.

The sections underwent a process of incubation in 0.1% trypsin for 10 min at 37 °C and then in 1 N HCl for 30 min at 56 °C, which served for DNA denaturation. Subsequently, these slices designated for immunohistochemical evaluation were rinsed with phosphate-buffered saline (PBS). The sections were pre-incubated in 0.3% hydrogen peroxide in PBS for 30 min to neutralize endogenous peroxidases. Additionally, to block non-specific protein binding, the sections were treated with PBS composed of 10% normal horse serum (Vector Laboratories, Burlingame, CA, USA) for 30 min.

Following this, the tissue slices were subjected to primary antibody treatment at concentrations between 1:100 and 1:500 for two hours at ambient temperature and subsequently washed three times with PBS. Next, a treatment with secondary antibodies (biotinylated anti-mouse IgG, diluted 1:100; Vector Laboratories) was provided to the sections for one hour at room temperature, succeeded by three additional PBS rinses. They were then subjected to an avidin–biotin–peroxidase complex (ABC; Vector Laboratories) for an hour, followed by three more washes with PBS. To develop the peroxidase reaction, the sections were immersed in a 0.05 mol/L Tris–HCl solution (pH 7.6) containing 0.01% H_2_O_2_ and 0.05% 3,3′-diaminobenzidine (DAB; Sigma-Aldrich). Afterward, the sections underwent counterstaining with hematoxylin, mounting, and analysis with an Axiophot photomicroscope (Carl Zeiss, Oberkochen, Germany) paired with an AxioCam MRc5 camera (Carl Zeiss).

The histological analysis was conducted in a manner that was blind to the details of the study, including the grouping of subjects, to maintain objectivity and independence throughout the investigation. The examination of muscle sections was performed using an Axiophot photomicroscope (Carl Zeiss). For each experimental group, five fields were chosen at random for imaging. The entire cross-section of the muscle, obtained from digital images of sections stained for anti-myosin to detect type I and II muscle fibers (at 100× magnification), was analyzed. The cross-sectional area (CSA) of these anti-myosin-positive muscle fibers was measured using specialized image analysis software (AxioVision SE64; Carl Zeiss), from which the average value was calculated.

For the evaluation of VEGF, PECAM-1, PCNA, and BrdU immunostaining, photographs from 20 fields randomly chosen for each group were taken and analyzed with AxioVision SE64 software (Carl Zeiss). The counts of VEGF, PECAM-1, PCNA, and BrdU-positive nuclei, along with the total of the muscle fibers on each photograph, were determined (Figure 13 and Figure 14). The results for VEGF, PECAM-1, PCNA, and BrdU were expressed as the ratio of the number of positive cells or nuclei to every 1000 muscle fibers.

### 4.7. Western Blot Analysis

Calf tissue sections were preserved by incubating them for one minute on ice in a radioimmunoprecipitation assay buffer. The buffer composition included 1× PBS, 1% NP-40, 0.5% sodium deoxycholate, 0.1% sodium dodecyl sulfate (SDS), 10 µg/mL of phenylmethanesulfonylfluoride, and a protease inhibitor cocktail tablet. Subsequently, the obtained homogeneous samples were subjected to centrifugation at 10,000× *g* for 10 min at 4 °C and stored at −70 °C for Western blot analysis. The protein content in the supernatants was measured using a BCA assay kit (Thermo Scientific Inc., Waltham, MA, USA) according to the detailed guidelines provided by the manufacturer.

For Western blot analysis, SDS–polyacrylamide gel electrophoresis with NuPAGE 4–12% bis–Tris gels (Invitrogen, Waltham, MA, USA) for protein samples weighing 40 µg each was carried out, and separated proteins were then transferred onto polyvinylidene difluoride (PVDF) membranes (GE Healthcare Life Sciences, Amersham, Bucks, Germany). Following that, the membranes were blocked using a casein-based blocking buffer (Sigma-Aldrich, St. Louis, MO, USA) before the application of specific antibodies. The membranes were then cleansed with PBS and Tween-20 and infested with an HRP-linked anti-mouse IgG whole antibody (sc-2005; Santa Cruz Biotechnology, Santa Cruz, CA, USA) at a concentration of 1:5000. Protein bands were made visible through improved chemiluminescence (Promega Corp., Madison, WI, USA), with an anti-β-actin antibody (A2228; Sigma-Aldrich, St. Louis, MO, USA) serving as the loading control. The primary antibodies targeted were PCNA (1:500, sc-56; Santa Cruz Biotechnology, Santa Cruz, CA, USA), VEGF (1:500, sc-7269; Santa Cruz Biotechnology, Santa Cruz, CA, USA), TNF-α (tumor necrosis factor α) (1:500, SC-52746; Santa Cruz Biotechnology, Santa Cruz, CA, USA), p-p38 (1:500, #9211, Cell signaling, Abcam, Cambridge, UK), p38 (1:500, #8690, Cell signaling), p-Akt (1:500, #9271, Cell signaling), Akt (1:500, #9272, Cell signaling), IL-1α (interleukin 1α) (1:500, ab239517; Abcam, Cambridge, UK), and IL-1β (interleukin 1β) (1:500, ab1832P; Merck Millipore, Burlington, MA, USA).

## 5. Statistical Analysis

A preliminary power analysis, informed by pilot study findings, established that a sample size of 24 subjects would achieve a statistical power of 0.8 with a significance threshold of 0.05. Statistical evaluation was conducted using SPSS software, version 22.0, for Windows (SPSS Inc., Chicago, IL, USA). Alongside basic descriptive statistics (averages and standard deviations), the analysis of variance (ANOVA) technique was employed to explore differences both within and between groups. If the ANOVA results revealed significant variances among groups, Tukey’s test was utilized for further post hoc comparisons. Average values are reported alongside 95% confidence intervals, and all the findings are presented as the means ± standard deviations. The statistical significance was predefined at *p* < 0.05. Following the study, a post hoc power analysis was carried out, indicating an achieved power exceeding 0.95.

## 6. Conclusions

In conclusion, Melittin, a major peptide found in honeybee venom, promotes muscle atrophy recovery by promoting the regeneration of muscle fibers in atrophied muscle in rabbits. To date, there has been scarce prior research comparing the regenerative effects of different Melittin concentrations on atrophied skeletal muscles. Therefore, this study sheds light on the superior efficacy of relatively high-dose Melittin over low-dose Melittin therapy in treating skeletal muscle atrophy provoked by immobilization casting. Melittin has the potential to serve as an effective therapy for recovery from muscle atrophy facilitated by chronic immobilization. Further research is required to assess the effects of Melittin at various dosages, frequency, and duration.

## Figures and Tables

**Figure 1 ijms-25-05035-f001:**
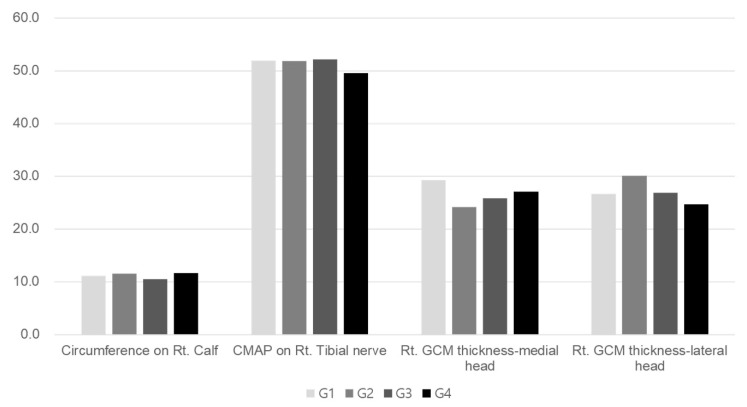
Comparison of atrophic change in clinical parameters among the four groups. There were no significant differences among the groups. *p* < 0.05, one-way ANOVA.

**Figure 2 ijms-25-05035-f002:**
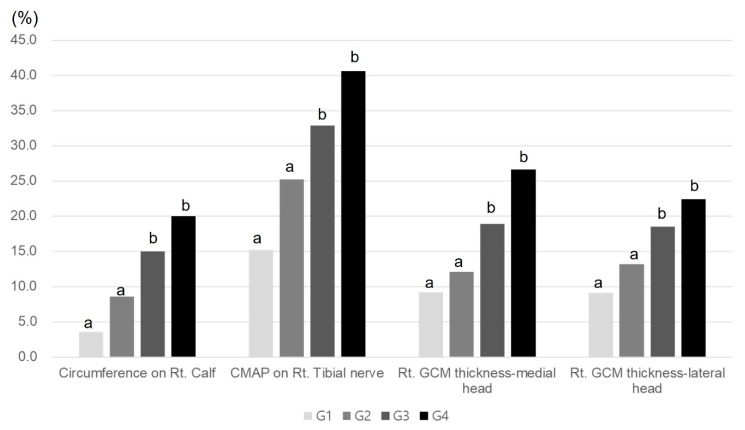
Comparison of regenerative effect of clinical parameters among the four groups. ^a,b^ Different letters on the bar mean the results were significantly different at *p* < 0.05 upon Tukey’s post hoc test among groups. GCM, gastrocnemius; CMAP, compound muscle action potential.

**Figure 3 ijms-25-05035-f003:**
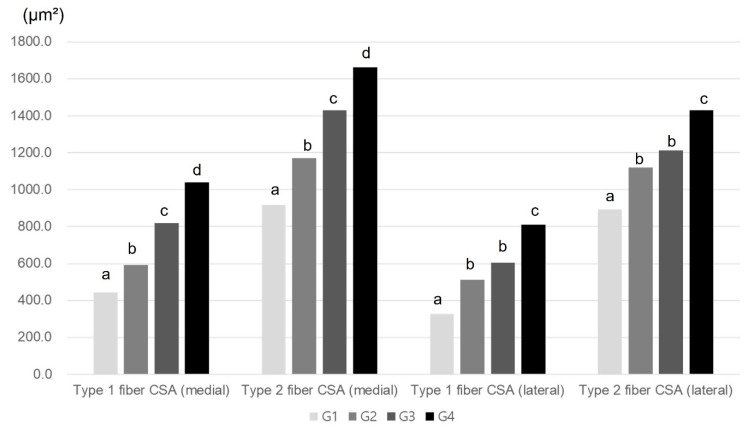
Comparison of mean CSA of GCM muscle fibers among the four groups. ^a–d^ Different letters on the bar mean the results were significantly different at *p* < 0.05 upon Tukey’s post hoc test among groups. GCM, gastrocnemius; CSA, cross-sectional area.

**Figure 4 ijms-25-05035-f004:**
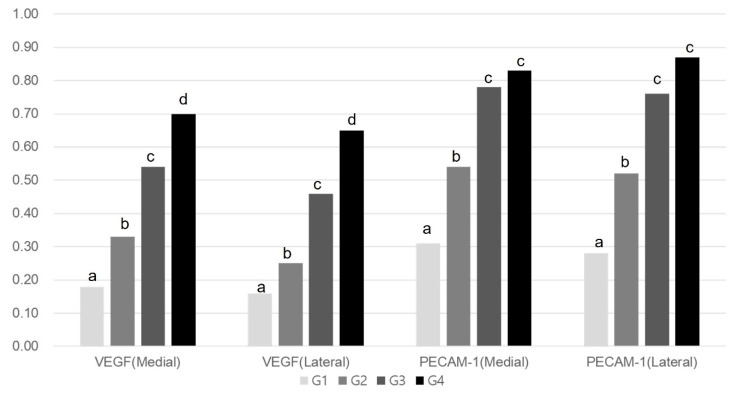
Comparison of VEGF and PECAM-1 ratios of the medial and lateral GCM muscle fibers among the four groups. ^a–d^ Different letters on the bar mean the results were significantly different at *p* < 0.05 upon Tukey’s post hoc test among groups. GCM, gastrocnemius; VEGF, vascular endothelial growth factor; PECAM-1, platelet endothelial cell adhesion molecule-1.

**Figure 5 ijms-25-05035-f005:**
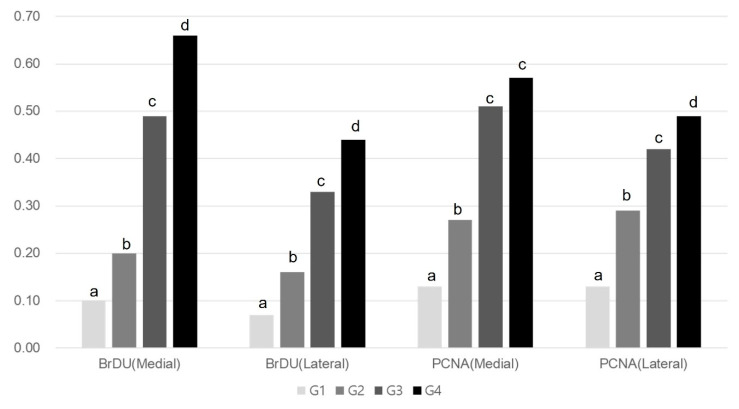
Comparison of BrdU and PCNA ratios of the medial and lateral GCM muscle fibers among the four groups. ^a–d^ Different letters on the bar mean the results were significantly different at *p* < 0.05 upon Tukey’s post hoc test among groups. GCM, gastrocnemius; BrdU, Bromodeoxyuridine; PCNA, proliferating cell nuclear antigen.

**Figure 6 ijms-25-05035-f006:**
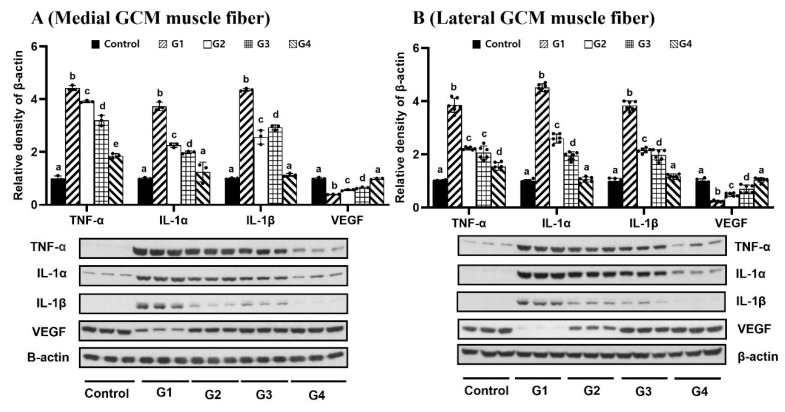
Results of tissue Western blotting show relative expression levels of tumor necrosis factor a (TNF-α), interleukin 1a (IL-1a), interleukin 1b (IL-1b), and vascular endothelial growth factor (VEGF) in (**A**) (medial GCM muscle fiber) and (**B**) (lateral GCM muscle fiber). ^a–e^ Different letters on the bar mean the results were significantly different at *p* < 0.05 upon post hoc testing among groups.

**Figure 7 ijms-25-05035-f007:**
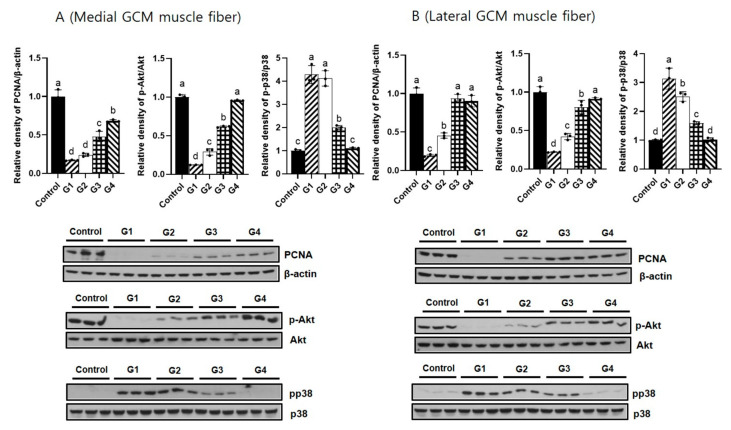
Results of tissue Western blotting show relative expression levels of proliferating cell nuclear antigen (PCNA), p38 mitogen-activated protein kinases (p38 MAPK), and Akt in (**A**) (medial GCM muscle fiber) and (**B**) (lateral GCM muscle fiber). ^a–d^ Different letters on the bar mean the results were significantly different at *p* < 0.05 upon post hoc testing among groups.

**Figure 8 ijms-25-05035-f008:**
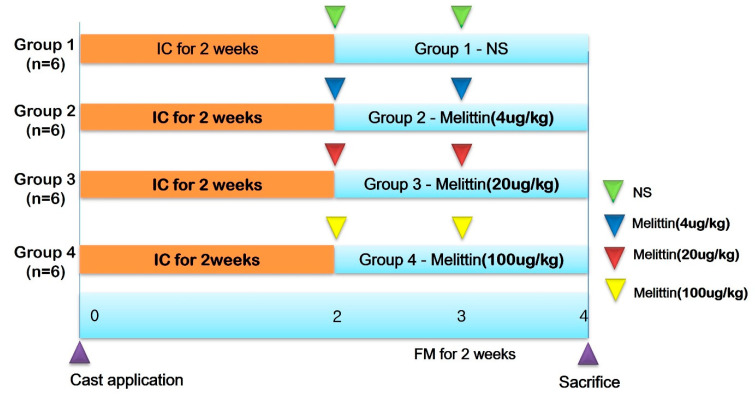
The study timeline involved 24 rabbits randomly assigned into four groups. Group 1 (G1) underwent immobilization by casting (IC) for two weeks, followed by daily injections of 0.2 mL normal saline for two weeks after cast removal (CR). Group 2 (G2) also experienced two weeks of IC, followed by daily injections of Melittin at a dose of 4 ug/kg for two weeks post-CR. Group 3 (G3) underwent the same initial two weeks of IC, followed by daily injections of Melittin at a dose of 20 ug/kg for two weeks after CR. Group 4 (G4) was subjected to two weeks of IC, followed by daily injections of Melittin at a dose of 100 ug/kg for two weeks post-CR. Definitions: IC, immobilized by cast; NS, normal saline; CR, cast removal.

**Figure 9 ijms-25-05035-f009:**
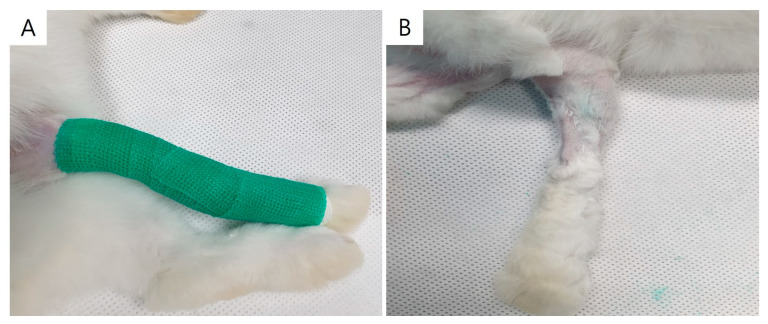
Immobilization of rabbit’s right lower limb using cast (**A**). Atrophied calf muscle in cast-immobilized rabbit model (**B**).

**Figure 10 ijms-25-05035-f010:**
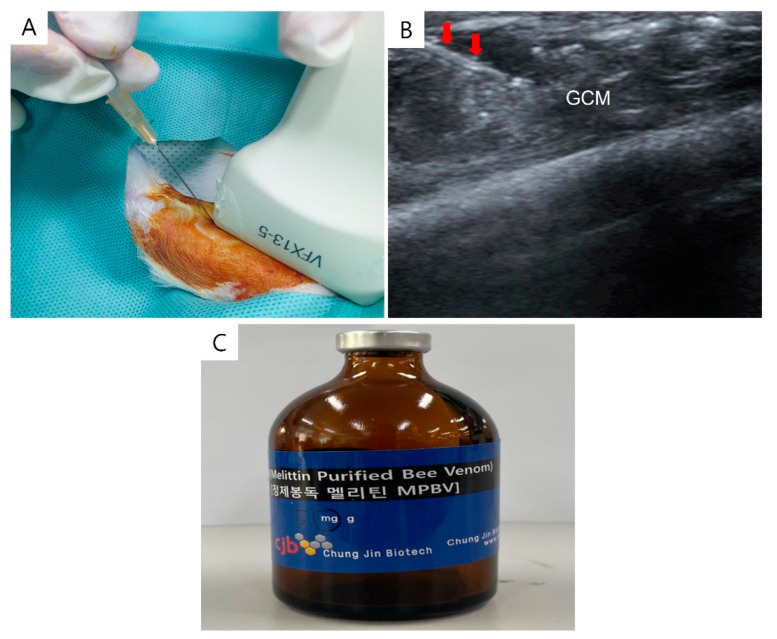
The longitudinal ultrasound image displays the process of injecting Melittin into a rabbit’s right gastrocnemius muscle. This procedure is guided by ultrasound, with the needle indicated by red arrows (**A**). The image also shows the gastrocnemius muscle (**B**) and the Melittin (**C**). GCM refers to the gastrocnemius muscle.

**Figure 11 ijms-25-05035-f011:**
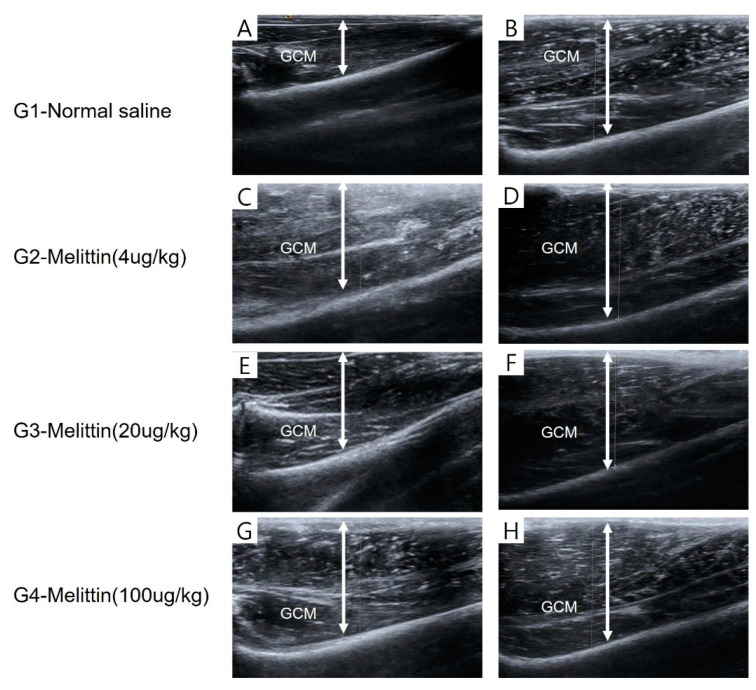
The thickness of the GCM (gastrocnemius muscle) was measured using ultrasound, defined as the distance between the superficial and deep aponeuroses of the GCM muscle, indicated by up–down arrows. The provided images are representative longitudinal sonograms of the right (**A**,**C**,**E**,**G**) and left (**B**,**D**,**F**,**H**) GCM muscles.

**Figure 12 ijms-25-05035-f012:**
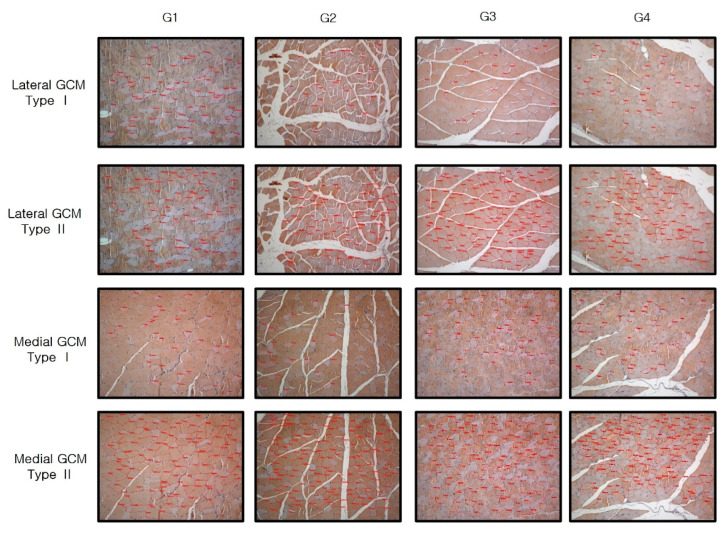
Immunohistochemical analysis of GCM (gastrocnemius muscle) fibers was conducted among the four groups, focusing on immobilized GCM muscles stained with monoclonal anti-myosin type II antibodies. The cross-sectional areas of type I and type II gastrocnemius muscle fibers, highlighted by red circles, were measured using an image morphometry program.

**Figure 13 ijms-25-05035-f013:**
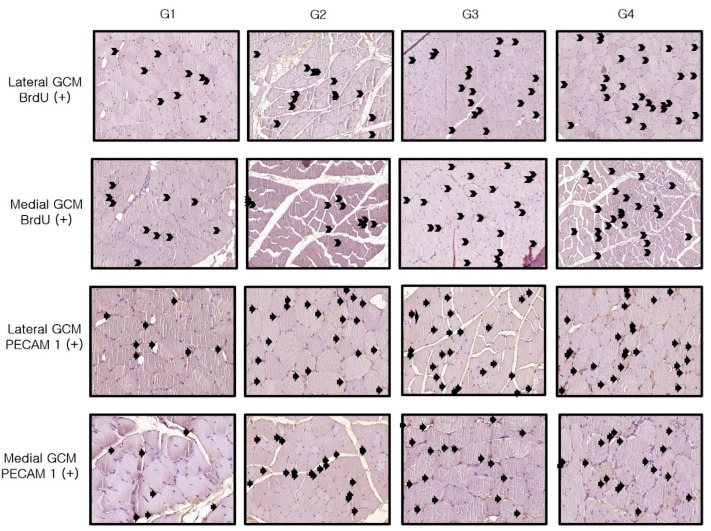
Immunohistochemical analysis of GCM (gastrocnemius muscle) fibers was conducted across the four groups, focusing specifically on immobilized GCM muscles stained with anti-BrdU and anti-PECAM-1 antibodies. Cells or nuclei positive for BrdU and PECAM-1, indicated by arrows, were counted along with the total number of muscle fibers within each image.

**Figure 14 ijms-25-05035-f014:**
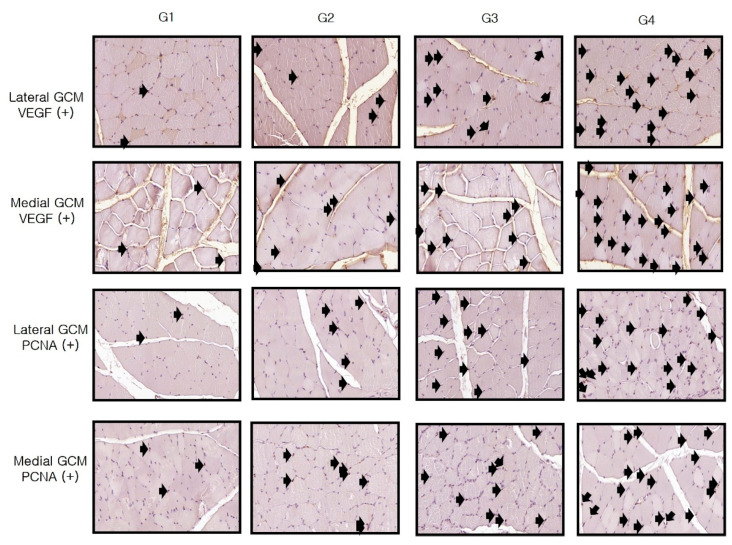
Immunohistochemical analysis of GCM (gastrocnemius muscle) fibers across the four groups involved examining immobilized GCM muscles stained with anti-VEGF and anti-PCNA antibodies. The number of cells or nuclei that were positive for VEGF and PCNA, identified by arrows, was counted, along with the total number of muscle fibers present in each image.

**Table 1 ijms-25-05035-t001:** Atrophic change in clinical parameters among the four groups.

Atrophic Change (%)				
Groups	Circumference on Rt Calf (cm)	CMAP on Rt. Tibial Nerve (mV)	Rt. GCM Thickness (mm)	
			Medial	Lateral
G1	11.1 ± 3.7	51.9 ± 7.9	29.2 ± 5.8	26.6 ± 5.4
G2	11.5 ± 4.6	51.8 ± 2.3	24.1 ± 11.7	30.1 ± 7.0
G3	10.5 ± 5.6	52.2 ± 3.0	25.8 ± 6.5	26.8 ± 4.3
G4	11.6 ± 6.0	49.5 ± 4.2	27.0 ± 7.1	24.1 ± 6.2

Values are presented as the mean ± standard deviation. G1-IC for 2 weeks and 0.2 mL normal saline injection for 2 weeks after CR; G2-IC for 2 weeks and Melittin (4 ug/kg) injection for 2 weeks after CR; G3-IC for 2 weeks and Melittin (20 ug/kg) injection after CR; G4-IC for 2 weeks and Melittin (100 ug/kg) injection for 2 weeks after CR. IC, immobilized by cast; CR, cast removal; GCM, gastrocnemius; CMAP, compound muscle action potential.

**Table 2 ijms-25-05035-t002:** Comparison of regenerative effect of clinical parameters among the four groups.

Regenerative Change (%)				
Groups	Circumference on Rt Calf (cm)	CMAP on Rt. Tibial Nerve (mV)	Rt. GCM Thickness (mm)	
			Medial	Lateral
G1	3.6 ± 2.0 ^a^	15.2 ± 8.4 ^a^	9.2 ± 5.0 ^a^	9.1 ± 13.2 ^a^
G2	8.6 ± 3.6 ^a^	25.2 ± 5.8 ^a^	12.1 ± 4.5 ^a^	13.2 ± 5.5 ^a^
G3	15.0 ± 5.4 ^b^	32.9 ± 5.9 ^b^	18.9 ± 2.9 ^b^	18.5 ± 5.8 ^b^
G4	20.0 ± 3.4 ^b^	40.6 ± 9.0 ^b^	26.6 ± 8.7 ^b^	22.4 ± 4.7 ^b^

Values are presented as the mean ± standard deviation. G1-IC for 2 weeks and 0.2 mL normal saline injection for 2 weeks after CR; G2-IC for 2 weeks and Melittin (4 ug/kg) injection for 2 weeks after CR; G3-IC for 2 weeks and Melittin (20 ug/kg) injection after CR; G4-IC for 2 weeks and Melittin (100 ug/kg) injection for 2 weeks after CR. ^a,b^ Different letters on the bar mean the results were significantly different at *p* < 0.05 upon Tukey’s post hoc test among groups. IC, immobilized by cast; CR, cast removal; GCM, gastrocnemius; CMAP, compound muscle action potential.

**Table 3 ijms-25-05035-t003:** Comparison of immunohistochemical findings in gastrocnemius muscle fiber among the four groups.

Immunohistochemical Findings	Group 1	Group 2	Group 3	Group 4
Medial GCM Type I fiber CSA (µm^2^) Type II fiber CSA (µm^2^)	443.51 ± 71.65 ^a^ 916.46 ± 113.81 ^a^	593.36 ± 127.54 ^b^ 1171.83 ± 149.07 ^b^	819.99 ± 89.63 ^c^ 1430.85 ± 109.31 ^c^	1039.9 ± 158.82 ^d^ 1661.77 ± 157.1 ^d^
BrdU ratio	0.1 ± 0.03 ^a^	0.2 ± 0.03 ^b^	0.49 ± 0.13 ^c^	0.66 ± 0.14 ^d^
PECAM-1 ratio	0.31 ± 0.11 ^a^	0.54 ± 0.11 ^b^	0.78 ± 0.17 ^c^	0.83 ± 0.12 ^c^
VEGF ratio PCNA ratio	0.18 ± 0.05 ^a^ 0.13 ± 0.05 ^a^	0.33 ± 0.08 ^b^ 0.27 ± 0.07 ^b^	0.54 ± 0.08 ^c^ 0.51 ± 0.08 ^c^	0.7 ± 0.17 ^d^ 0.57 ± 0.13 ^c^
Lateral GCM Type I fiber CSA (µm^2^) Type II fiber CSA (µm^2^)	327.77 ± 64.66 ^a^ 892.59 ± 66.4 ^a^	513.22 ± 89.83 ^b^ 1119.3 ± 106.96 ^b^	603.76 ± 57.67 ^b^ 1211.12 ± 91.44 ^b^	811.24 ± 143.69 ^c^ 1430.9 ± 207.23 ^c^
BrdU ratio	0.07 ± 0.02 ^a^	0.16 ± 0.03 ^b^	0.33 ± 0.09 ^c^	0.44 ± 0.10 ^d^
PECAM-1 ratio	0.28 ± 0.07 ^a^	0.52 ± 0.17 ^b^	0.76 ± 0.16 ^c^	0.87 ± 0.18 ^c^
VEGF ratio PCNA ratio	0.16 ± 0.02 ^a^ 0.13 ± 0.04 ^a^	0.25 ± 0.07 ^b^ 0.29 ± 0.06 ^b^	0.46 ± 0.11 ^c^ 0.42 ± 0.1 ^c^	0.65 ± 0.09 ^d^ 0.49 ± 0.1 ^d^

Values are presented as the mean ± standard deviation. G1-IC for 2 weeks and 0.2 mL normal saline injection for 2 weeks after CR; G2-IC for 2 weeks and Melittin (4 ug/kg) injection for 2 weeks after CR; G3-IC for 2 weeks and Melittin (20 ug/kg) injection after CR; G4-IC for 2 weeks and Melittin (100 ug/kg) injection for 2 weeks after CR. ^a–d^ Different letters on the bar mean the results were significantly different at *p* < 0.05 upon Tukey’s post hoc test among groups. GCM, gastrocnemius; CSA, cross sectional area; BrdU, Bromodeoxyuridine; PECAM-1, platelet endothelial cell adhesion molecule-1; VEGF, vascular endothelial growth factor; PCNA, proliferating cell nuclear antigen.

## Data Availability

All data produced or assessed during this study are included in the article.

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
