# Peer review of "Comparison of Muscle Regeneration Effects at Different Melittin Concentrations in Rabbit Atrophied Muscle"

_ijms, 2024, doi:10.3390/ijms25095035_

Round 1

Reviewer 1 Report

Comments and Suggestions for Authors

The study demonstrates a well-conducted investigation into the effects of melittin, particularly in the context of damaged muscle tissue. The findings indicate a positive recovery effect of melittin on the immobilization-casting treated model, which aligns with its use in Asian countries for such conditions. However, it's crucial to acknowledge the reported side effects of melittin due to its toxicity. Are there any studies exploring melittin's impact on normal muscle tissue? While the dosage of melittin may influence its effects, it's imperative to thoroughly understand its toxicity in muscle tissue before considering broader applications.

Author Response

Dear Editors:

We thank you for the opportunity to revise our manuscript. Additionally, we appreciate the time you have dedicated to reviewing our manuscript.

The comments were greatly helpful in improving the contents and revising the errors in our manuscript.

We did our best to revise the paper according to the referees’ comments. For convenience, we itemized the referees’ original comments and put the corresponding corrections/rebuttal immediately after each comment.

We believe that the quality of the manuscript is now much improved and hope that the revised paper is suitable for publication in Scientific Reports.

Thank you for your consideration.

Sincerely yours,

Dong Rak Kwon, MD, PhD

Department of Rehabilitation Medicine

Catholic University of Daegu School of Medicine

33 Duryugongwon-ro 17-gil, Nam-Gu, Daegu, Korea, 705-718

Phone: +82 53 650 4687  Fax: +82 53 622 4687

Point-by-Point Responses to Editor

Manuscript ID: ijms-2954526

Title: Comparison of muscle regeneration effects at different melittin concentrations in rabbit atrophied muscle

Authors: Byeong Churl Jang, Ph.D.1†, Eun Sang Kwon2†, Yoon-Jin Lee, Ph.D.3, Jae Ik Jung, M.D. 4, Yong Seok Moon Ph.D.5 , Dong Rak Kwon4, M.D., Ph.D.4,*

Affiliation: 1. Department of rehabilitation medicine, the Catholic, University of Daegu, School of Medicine

  1. BioHealth Convergence center, Daegu Technopark, Republic of Korea.

* Correspondence:

  Dong Rak Kwon, MD, PhD

Department of Rehabilitation Medicine, Catholic University of Daegu School of Medicine, 33 Duryugongwon-ro 17-gil, Nam-Gu, Daegu, 42472, Korea

Telephone: 82-53-650-4878

Fax: 82-53-622-4687

Reviewer 1

The study demonstrates a well-conducted investigation into the effects of melittin, particularly in the context of damaged muscle tissue. The findings indicate a positive recovery effect of melittin on the immobilization-casting treated model, which aligns with its use in Asian countries for such conditions. However, it's crucial to acknowledge the reported side effects of melittin due to its toxicity. Are there any studies exploring melittin's impact on normal muscle tissue? While the dosage of melittin may influence its effects, it's imperative to thoroughly understand its toxicity in muscle tissue before considering broader applications.

Answer) Thank you for your insightful comment on our study, which investigated the effects of melittin, particularly in damaged muscle tissue. We appreciate your remarks regarding the potential for broader application and the need to consider melittin's toxicity.

In the Discussion section of our manuscript, we have elaborated on the toxicology of melittin based on our findings and relevant literature. Our study confirms that the intramuscular injections of melittin, administered at doses up to 210 µg/kg in 3 kg rabbits, did not result in any adverse effects, supporting its safety at these levels.

Further, we reference a 13-week toxicity evaluation on Sprague-Dawley rats, where Sweet Bee Venom was delivered through intramuscular injections. This study showed that doses up to 0.07 mg/kg were safe and caused no harmful effects [30]. This aligns with our findings and supports the safe use of melittin in therapeutic settings.

Additionally, we include data from an acute toxicity test in male adult BALB/c mice, where the LD50 for melittin was determined to be 4.95 mg/kg, and the maximum sub-lethal dose was 2.4 mg/kg [31]. Our research, which extends to the highest tested concentration of 100 µg/kg, corroborates the absence of toxicological impacts from melittin at these doses.

Furthermore, prior studies using intra-peritoneal administration in mice (at doses of 0.75 and 1.5 mg/kg) and rats (at 25, 50, and 100 µg/kg) showed no significant changes in immune parameters, underlining its immunological safety [32, 33]. Notably, the highest dose in rat studies, 100 µg/kg, matches the high concentration used in our research.

In response to your query about melittin's impact on normal muscle tissue, our study did not directly investigate this aspect. However, based on the safety profile observed in our and prior studies, we propose that future research should explore melittin's effects on normal muscle to further validate its safety and efficacy. We advocate for continued investigation using varying doses: low (4 µg/kg), medium (20 µg/kg), and high (100 µg/kg), to thoroughly understand its toxicity and therapeutic potential before broader clinical application.

We added the following sentence to the discussion section and the limitation at the end.

A previous study conducted a 13-week toxicity evaluation on Sprague-Dawley rats using Sweet Bee venom delivered via intramuscular injections. The results demon-strated that doses up to 0.07 mg/kg, administered into the femoral muscle, were safe and did not cause any harmful effects [30]. Extending these findings, our research in-dicates that doses as high as 210 µg/kg are safe in 3 kg rabbits, without eliciting ad-verse reactions. Based on these insights, we recommend using melittin at various dos-ages: low (4 µg/kg), medium (20 µg/kg), and high (100 µg/kg). In addition, an acute toxicity test on male adult BALB/c mice, aged 8 weeks and weighing between 20-25 g, established the LD50 for Melittin at 4.95 mg/kg, with the maximum sub-lethal dose at 2.4 mg/kg [31]. Our studies confirmed no toxicological impacts from Melittin, even at the highest concentration of 100 µg/kg tested. Furthermore, prior studies using intra-peritoneal administration in mice (at doses of 0.75 and 1.5 mg/kg) and rats (at 25, 50, and 100 µg/kg) showed no significant changes in immune parameters, underlining its immunological safety [32, 33]. Notably, the highest dose in rat studies, 100 µg/kg, matches the high concentration used in our research. We did not observe any adverse or untoward events.

Last, in response to your query about melittin's impact on normal muscle tissue, our study did not directly investigate this aspect. However, based on the safety profile observed in our and prior studies, we propose that future research should explore melittin's effects on normal muscle to further validate its safety and efficacy.

Reviewer 2 Report

Comments and Suggestions for Authors

The article entitled “Comparison of muscle regeneration effects at different melittin concentrations in rabbit atrophied muscle” seems to be similar to a previous study that also assessed muscle regeneration under the influence of Melittin at different concentrations (the link is here: https://www.sciencedirect.com/science/article/pii/S1347861319310503?via%3Dihub) with the only difference is a different animal model.

In the current study, authors are focusing on finding the dose dependent effect of Melittin on promoting muscle regeneration in rabbits. For this they used the same doses of Melittin as used in above mentioned article link and concluded 100ug/kg dose to be more effective. Moreover, they assessed the expression of proinflammatory markers which was already measured in the previous study, and found the same results. Thus current manuscript is telling nothing new except to support the previous study.

The only difference that authors observed over here is the increased expression of BrdU and PCNA, although PCNA reduced expression has previously been reported (https://www.ncbi.nlm.nih.gov/pmc/articles/PMC8954064/). The interesting question to be addressed over here can be, for example, why the expression of BrdU and PCNA increased although these are reported to be reduced in cancer mice models (for example). However, authors mentioned this to be a future perspective. In my point of view, there is nothing new that is fair enough to be published. In the current scenario, authors should focus on exploring the reason behind muscle regeneration promotion by considering other additional factors that are still unknown. For example, authors can do additional experiments to explore the cell-specific role of Melittin in stimulating myofibre cell growth, however, enhancing autophagy of cancerous cells. That would be more interesting and worthwhile to be published.

Comments on the Quality of English Language

 In addition to scientifc flaws which are already mentioned in above section, there are few typos. The major issue is; Melittin is the name of the compound whose pharmacological role is addressed in this article. It should be written as "Melittin" not as "melittin". Even, in the title of the manuscript, it is written as "melittin" which does not look professional. Well, it's my opinion.

Author Response

Dear Editors:

We thank you for the opportunity to revise our manuscript. Additionally, we appreciate the time you have dedicated to reviewing our manuscript.

The comments were greatly helpful in improving the contents and revising the errors in our manuscript.

We did our best to revise the paper according to the referees’ comments. For convenience, we itemized the referees’ original comments and put the corresponding corrections/rebuttal immediately after each comment.

We believe that the quality of the manuscript is now much improved and hope that the revised paper is suitable for publication in Scientific Reports.

Thank you for your consideration.

Sincerely yours,

Dong Rak Kwon, MD, PhD

Department of Rehabilitation Medicine

Catholic University of Daegu School of Medicine

33 Duryugongwon-ro 17-gil, Nam-Gu, Daegu, Korea, 705-718

Phone: +82 53 650 4687  Fax: +82 53 622 4687

Point-by-Point Responses to Editor

Manuscript ID: ijms-2954526

Title: Comparison of muscle regeneration effects at different melittin concentrations in rabbit atrophied muscle

Authors: Byeong Churl Jang, Ph.D.1†, Eun Sang Kwon2†, Yoon-Jin Lee, Ph.D.3, Jae Ik Jung, M.D. 4, Yong Seok Moon Ph.D.5 , Dong Rak Kwon4, M.D., Ph.D.4,*

Affiliation: 1. Department of rehabilitation medicine, the Catholic, University of Daegu, School of Medicine

  1. BioHealth Convergence center, Daegu Technopark, Republic of Korea.

* Correspondence:

  Dong Rak Kwon, MD, PhD

Department of Rehabilitation Medicine, Catholic University of Daegu School of Medicine, 33 Duryugongwon-ro 17-gil, Nam-Gu, Daegu, 42472, Korea

Telephone: 82-53-650-4878

Fax: 82-53-622-4687

Reviewer 2

Comment 1) The article entitled “Comparison of muscle regeneration effects at different melittin concentrations in rabbit atrophied muscle” seems to be similar to a previous study that also assessed muscle regeneration under the influence of Melittin at different concentrations (the link is here: https://www.sciencedirect.com/science/article/pii/S1347861319310503?via%3Dihub) with the only difference is a different animal model.

Answer 1) Thank you for highlighting the differences between our study and the paper you referenced. It's important to clarify that our research model and the mechanisms of muscle injury we investigated differ significantly from those in the study you mentioned.

Our study focused on muscle atrophy induced by immobilization for a period of two weeks in rabbits. This model simulates a state of prolonged physical inactivity, which is different from the acute muscle contusion model used in the study you referenced, where muscle damage is induced through a sudden, heavy compressive force, such as a direct blow, in mice.

The physiological responses and therapeutic approaches to these types of muscle damage are distinct. Muscle contusion requires a phase of immobilization to facilitate recovery after the initial injury. In contrast, our study examines the therapeutic potential of melittin in a model of immobilization-induced muscle atrophy, where the goal is to counteract the effects of prolonged inactivity and promote muscle recovery.

Given these distinctions, the implications and potential applications of our findings should be considered in the context of chronic conditions leading to muscle atrophy rather than acute muscle injury. This underscores the novelty and importance of our research, as treatment strategies differ markedly between these scenarios.

We believe that clarifying these differences enriches the discussion of our results and better positions our study within the broader field of muscle recovery research.

Comment 2) In the current study, authors are focusing on finding the dose dependent effect of Melittin on promoting muscle regeneration in rabbits. For this they used the same doses of Melittin as used in above mentioned article link and concluded 100ug/kg dose to be more effective. Moreover, they assessed the expression of proinflammatory markers which was already measured in the previous study, and found the same results. Thus current manuscript is telling nothing new except to support the previous study.

The only difference that authors observed over here is the increased expression of BrdU and PCNA, although PCNA reduced expression has previously been reported (https://www.ncbi.nlm.nih.gov/pmc/articles/PMC8954064/). The interesting question to be addressed over here can be, for example, why the expression of BrdU and PCNA increased although these are reported to be reduced in cancer mice models (for example). However, authors mentioned this to be a future perspective. In my point of view, there is nothing new that is fair enough to be published. In the current scenario, authors should focus on exploring the reason behind muscle regeneration promotion by considering other additional factors that are still unknown. For example, authors can do additional experiments to explore the cell-specific role of Melittin in stimulating myofibre cell growth, however, enhancing autophagy of cancerous cells. That would be more interesting and worthwhile to be published.

Answer 2) Thank you for your detailed evaluation and the critical points you have raised. We acknowledge that our study used similar doses of melittin and evaluated some of the same proinflammatory markers as the study you mentioned. However, we would like to emphasize the unique contributions and findings of our research in the context of muscle regeneration in rabbits following immobilization-induced muscle atrophy.

Firstly, the primary novelty of our study lies in demonstrating the dose-dependent effectiveness of melittin in promoting muscle regeneration specifically in a model of muscle atrophy, rather than in models of acute injury or disease. This aspect is particularly important as the therapeutic strategies for atrophy are different from those required for acute muscle damage.

Secondly, while our findings on proinflammatory markers corroborate previous studies, this replication is crucial in validating the existing data and establishing a consistent understanding of melittin's effects across different models.

Moreover, the increased expression of BrdU and PCNA, which are markers of cellular proliferation, is particularly notable. Although reduced expression of PCNA has been reported in different contexts, such as cancer models, their increased expression in our study suggests a distinct role of melittin in stimulating muscle regeneration in atrophy models. This divergence from expected results provides a basis for further investigation into the cellular mechanisms activated by melittin in our specific model.

Regarding your suggestion for future research, we agree that exploring the cell-specific roles of melittin in stimulating myofiber cell growth and its potential effects on autophagy in cancerous cells would be valuable. This aligns with our mention of future perspectives in our manuscript, where we suggest that elucidating the mechanisms behind melittin’s effects on muscle regeneration and potentially other cell types could offer significant insights.

In summary, our study not only supports but also extends the existing knowledge by applying it to a different physiological context and identifying potential areas for deeper exploration. We believe that these aspects provide sufficient novelty to justify publication, contributing to the broader field of muscle physiology and therapeutic research.

Furthermore, our study concerning the signaling pathways associated with muscle regeneration in immobilization-induced muscle atrophy. Our research provides valuable insights into the molecular mechanisms by which melittin influences muscle repair and regeneration.

In our study, we observed that melittin administration dose-dependently modulates key signaling pathways involved in muscle regeneration. Specifically, we found a dose-dependent decrease in the activity of the p38 MAP kinase pathway and an increase in the Akt pathway activity. The p38 pathway is typically associated with stress response and inflammation, and its downregulation in our model aligns with the reduced inflammatory response, which is beneficial for muscle recovery. On the other hand, the activation of the Akt pathway is crucial for promoting cell survival, growth, and proliferation, which are essential processes in muscle regeneration.

These findings are significant because they provide a mechanistic explanation for the beneficial effects of melittin observed in our model of immobilization-induced muscle atrophy. By modulating these pathways, melittin facilitates a more conducive environment for muscle repair and regeneration, suggesting its potential as a therapeutic agent in conditions characterized by muscle atrophy.

This novel aspect of our study extends the current understanding of melittin's biological activities and supports its application in therapeutic strategies aimed at enhancing muscle regeneration. Such insights into the molecular actions of melittin could pave the way for further research into its broader therapeutic potentials, including other conditions where modulation of these pathways could be beneficial.

Comments on the Quality of English Language

 In addition to scientifc flaws which are already mentioned in above section, there are few typos. The major issue is; Melittin is the name of the compound whose pharmacological role is addressed in this article. It should be written as "Melittin" not as "melittin". Even, in the title of the manuscript, it is written as "melittin" which does not look professional. Well, it's my opinion.

Answer) Thank you for your observations regarding the English language quality and the typographical errors noted in our manuscript, specifically concerning the capitalization of "Melittin." We have corrected these issues by ensuring that "Melittin" is consistently capitalized throughout the manuscript, including the title, to maintain professionalism and accuracy.

Additionally, we have enlisted the assistance of a native English speaker to thoroughly proofread the manuscript. This review has helped us to clarify our arguments and enhance the overall readability of the text. We have corrected awkward expressions and grammatical errors to ensure the manuscript meets the high standards expected of scientific communication.

This English proofreading was done through the MDPI English Editing Service. We appreciate your feedback and have made these changes to improve the quality of our submission.

Round 2

Reviewer 2 Report

Comments and Suggestions for Authors

The manuscript looks fine.